# Wound Repair and Extremely Low Frequency-Electromagnetic Field: Insight from In Vitro Study and Potential Clinical Application

**DOI:** 10.3390/ijms22095037

**Published:** 2021-05-10

**Authors:** Giulio Gualdi, Erica Costantini, Marcella Reale, Paolo Amerio

**Affiliations:** 1Department of Medicine and Aging Science, Dermatology Clinic, University “G. D’Annunzio”, Via dei Vestini, 66100 Chieti, Italy; paolo.amerio@unich.it; 2Department of Medical, Oral and Biotechnological Sciences, University “G. D’Annunzio”, Via dei Vestini, 66100 Chieti, Italy; erica.costantini@unich.it (E.C.); mreale@unich.it (M.R.)

**Keywords:** ELF-EMF, wound, healing, fibroblasts, keratinocytes, non-healing wounds

## Abstract

Wound healing is a complex, staged process. It involves extensive communication between the different cellular constituents of various compartments of the skin and its extracellular matrix (ECM). Different signaling pathways are determined by a mutual influence on each other, resulting in a dynamic and complex crosstalk. It consists of various dynamic processes including a series of overlapping phases: hemostasis, inflammation response, new tissue formation, and tissue remodeling. Interruption or deregulation of one or more of these phases may lead to non-healing (chronic) wounds. The most important factor among local and systemic exogenous factors leading to a chronic wound is infection with a biofilm presence. In the last few years, an increasing number of reports have evaluated the effects of extremely low frequency (ELF) electromagnetic fields (EMFs) on tissue repair. Each experimental result comes from a single element of this complex process. An interaction between ELF-EMFs and healing has shown to effectively modulate inflammation, protease matrix rearrangement, neo-angiogenesis, senescence, stem-cell proliferation, and epithelialization. These effects are strictly related to the time of exposure, waveform, frequency, and amplitude. In this review, we focus on the effect of ELF-EMFs on different wound healing phases.

## 1. Introduction

Wound healing is a complex and well-regulated process controlled by extensive communication between cells, with a dynamic and complex crosstalk between different signaling pathways [1].

This process is composed of many phases, which include, consecutively, the hemostasis, inflammation response, new tissue formation, and tissue remodeling phases [2,3]. The hemostatic phase results from the immediate activation of platelets. These cells release molecules, such as a growth factor and cytokines, that prevent bleeding and initiate wound repair. The second step is inflammation, which is characterized by the afflux 24 to 48 h after injury of different immune cells, such as neutrophils, monocytes, and lymphocytes. These cells work together and in close coordination to prevent infection and to remove dead tissue [3,4]. Two to ten days following tissue injury, cellular proliferation, and migration of different cell types, such as fibroblasts, keratinocytes, and endothelial cells, new tissue formation occurs [5,6]. Fibroblasts are known as the main actors regulating the wound repair process and, in the presence of the wound microenvironment, migration, proliferation [6,7,8] and synthesis, and secretion of many factors, such as Matrix metalloproteinase-14 (MMP-14), basic fibroblast growth factor (bFGF), and fibroblast growth factor-9 (FGF-9), collagen homeostasis and angiogenesis increase [9,10].

Finally, in the re-modelling phase, two to three weeks after injury, fibroblasts differentiate into myofibroblasts [11], which produces an extracellular matrix leading to a mature scar [12]. The tissue remodeling process may last for a year or more. At this stage, all the processes started by injury will turn off through apoptosis of involved cells, fibroblasts, macrophages, and endothelial cells [10,11,12,13,14]. The wound will be repaired only if all these classic healing steps work correctly and in close coordination.

The process is highly efficient, but sometimes it can deviate from its physiological course, resulting in an ulcerative skin defect (chronic wound) or an excessive scar formation (hypertrophic scar or keloid). Chronic wound development may be common in various conditions including pressure, diabetes, venous pathology (venous, arterial, mixed, and vasculitis), trauma, and surgery, with significant morbidity and mortality risk [15,16] as well as impact for a healthy economy [17,18].

ELF-EMFs are non-ionizing, low-energy, electromagnetic fields capable of inducing several biological effects. Frequencies considered to be ELFs range from 3 Hz to 300 Hz. The study of the interaction between ELF-EMFs and the tissue is not always easy since different biological effects are related to EMFs’ time of exposure, waveform, frequency, amplitude, cell type, and cell status [19,20].

The ELF-EMFs are commonly produced by electrical devices, high tension electrical distribution networks, from residential and occupational sources, and by power lines. Low-frequency electric fields influence all systems characterized by charged particles as the human body. In fact, tiny electrical currents exist in the human body due to the chemical reactions that occur as part of normal bodily functions, even in the absence of external electric fields.

The interest in the biological interaction of ELF-EMFs with tissues has, nevertheless, increased due to their possible effect on human health as well as their potential therapeutic use. ELF-EMFs with frequencies less than 300 Hz do not have enough energy to break molecular bonds, nor cause DNA damage, ionization, or even to have thermal effects on cells and tissues [21]. Biological effects modulated by EMFs are very wide and include cell migration, proliferation and differentiation, cytokine and growth factors expression, and nitric oxide signaling alteration [20,21,22,23,24,25]. ELF-EMFs can interact with the chemical and biological processes modulating the physiological homeostasis, and, thus, can interact in wound healing.

Despite some studies reporting potential negative effects of ELF-EMFs, such as increased risk of childhood cancer, breast cancer, neoplastic development, neurodegenerative diseases, and in fertility, cardiovascular disorders, disease promotion, and progression [26,27,28,29,30,31,32,33], no convincing evidence was ever provided for a direct relationship between ELF-EMFs and disease development. In the last few years, an increasing number of reports have evaluated the effects of ELF-EMFs on tissue repair.

In this review, we aim to relate the in vitro knowledge with the potential clinical applications of low frequency fields in tissue repair.

## 2. EMF-ELF and Wound-Repair: Mechanism of Action

Despite the high ability of the innate reparative process, multiple cellular aspects of an individual’s injury response can be disturbed, compromising wound closure, and leading to chronic wound development. Recent studies have shed light on bioeffects induced by the EMF and how they might control tissue regeneration and wound healing, suggesting that EMF has a positive impact on all the different stages of healing. In fact, a promising novel strategy for treating the chronic wound may be the local delivery of ELF-EMFs to target resident cells to improve their ability in modulating immune responses and tissue healing.

### 2.1. ELF-EMFs and Hemostatic Phase

The initial phase of wound repair after injury is characterized by blood vessel damage and formation of a blood clot. The relationship between ELF-EMFs and platelets, as principal contributors to haemostasias and coagulation, has recently drawn interest. Platelets represent the inducers for the bleeding prevention mechanism and for the initiation of repair systems, supporting the recruitment of immune cells, cytokines, and growth factors necessary for early wound repair. A previous in vivo study reported contrasting results. Lai et al. observed no significant variation in platelet count after 100-μT ELF-EMFs exposure [34], while Liu et al. reported that exposure to ELF-EMFs increases the number of white blood cells (WBCs) and lymphocytes but decreases the mean platelet volume (MPV) levels at a bandwidth of 5 Hz to 32 KHz [35]. The mechanism of platelet activation is complex and required the increase of calcium levels, protein kinase C stimulation, and free oxygen radical generation.

ELF-EMFs may be generated with frequencies close to the resonant patterns of calcium (Ca^2+^), sodium (Na^+^), and other ions. Numerous studies reported that Ca^2+^ ions are the main target of ELF-EMFs [36]. Many studies have shown that voltage-dependent calcium channels may account for the biological effects of ELF-EMFs exposure. As an indirect proof of calcium role, it has also been shown that calcium channel blockers can greatly reduce the effects of 1 mT and 50-Hz exposure, and cause interference in cell differentiation and neurogenesis [37]. It is well documented that Ca^2+^ ions affect activity-dependent gene expression [38] and this effect is mediated by signaling pathways activating Ca^2^ -responsive DNA regulatory elements. Due to this direct cellular interaction, electromagnetic fields have been demonstrated to increase healing rates much faster than other therapies, as they may reach the deep tissue more quickly immediately after the insult [39] (Figure 1).

EMF quickly restores the balance between free radicals and antioxidants to stop the cascade of inflammatory progression and biochemical degradation in traumatized tissue [40]. Free radicals can be mitogenic or cytotoxic depending on levels, antioxidant system efficiency, and cell types. Previous studies have shown that reactive oxygen species (ROS), generated after brief exposure to ELF-EMFs, play a key role in cell proliferation as a possible initial cell event. Conversely, the continuous generation of ROS by long-term 50-60 Hz ELF-EMFs exposures can induce the accumulation of DNA damage and slow cell cycle progression [41,42]. ELF-EMFs have also been reported to up-regulate clusters of protective and restorative gene loci as well as down-regulate deregulatory and apoptotic gene loci [39].

### 2.2. ELF-EMFs and Inflammatory Phase

A large body of evidence on chronic wound tissue and fluids demonstrates an unstable competition between pro-inflammatory and anti-inflammatory signals that lead to the misbalanced environment favoring the development of chronic wounds [43,44].

It has also been shown that increased pro-inflammatory cellular infiltrates, composed largely of neutrophils and macrophages, contribute to delayed healing in chronic ulcers [45,46]. As a result, deregulation of several key pro-inflammatory cytokines, such as interleukin (IL)-1β and tumor necrosis factor-α (TNF-α), prolong the inflammatory phase and delay healing [44,47]. IL-1β and TNFα are increased in chronic wounds, and this increase has been shown to cause elevated levels of metalloproteinases that excessively degrade the local ECM and, thus, impair cell migration [48].

EMFs effects on the expression of cytokines have been mostly investigated with ex vivo and in vitro experiments on different cell types involved in tissue repair. Several reports have supported the anti-inflammatory effects of EMFs on tissue repair. Vianale et al. demonstrate that 50 Hz ELF-EMFs exposure may inhibit inflammatory processes by producing Regulated upon Activation, Normal T Cell Expressed and Presumably Secreted (RANTES), Macrophage chemotactic protein-1 (MCP-1), macrophage inflammatory protein (MIP)-1α and IL-8, and activation of nuclear factor kappa-light-chain-enhancer of activated B cells (NF-kB) inflammatory signaling pathways in keratinocytes in vitro [49].

The effect of ELF-EMFs on transition from a chronic pro-inflammatory to an anti-inflammatory state of the healing process was also reviewed by Pesce et al. [50]. In the epidermal wound healing process, ELF-EMFs exposure is reported to mediate keratinocyte proliferation, up-regulation of the Nitric Oxide Synthase (NOS) activities, and down-regulation of Cyclooxygenase-2 (COX-2) expression and Prostaglandin E2 (PGE-2) production, involved in the inflammatory response modulation [51].

The effects of ELF-EMFs on the inflammatory molecules are timing and cell type dependent. In fact, 50 Hz of exposure induced an early increase of IL-1β, IL-18, and TNFα production and secretion by keratinocytes and fibroblasts, while a later inhibition of inflammatory mediators led to healing, mirroring the physiological process of wound repair [52,53,54]. In accordance, in vivo and in vitro studies reported the influence of ELF-EMFs exposure on inflammatory state promotion, with increased production of pro-inflammatory cytokines, such as IL-6, IL-9, and TNF-α, and increased levels of ROS [55,56]. Furthermore, ex vivo studies reported the ability of ELF-EMFs exposure in activating the anti-inflammatory response by down-modulating pro-inflammatory cytokines, inducing IL-10, and transforming growth factor β (TGFβ), as a key antagonist of pro-inflammatory mediators [55,57], which enhanced the immune system response in the damaged areas and favored early wound healing. A clinical and experimental investigation showed that ELF-EMF speed up the switch from an inflammatory phase to a proliferative phase in the wound healing.

### 2.3. ELF-EMFs and Proliferative Phase

In the healing process, the proliferative phase is characterized by the activation of a wide array of cells: keratinocytes, fibroblasts, macrophages, and endothelial cells. This leads to complete wound healing, through matrix deposition and angiogenesis. This phase starts 12 h after the damage, with release of MMPs and migration stimulation, while new ECM proteins reconstitute the basement membrane.

MMPs, such as collagenase and gelatinases A and B, are more elevated in chronic wounds as compared to acute wounds [43]. ELF-EMFs were shown to upregulate MMP-9 release early with a physiologically late decrease [57]. Also, Wang et al., in a study on scleral fibroblasts, report that 0.2 mT of ELF-EMFs might act by increasing the expression of MMP-2 and reducing collagen I synthesis, which can be involved in pathological matrix remodeling [58].

This early up-regulation of MMPs activity and/or expression may represent a mechanism to promote the migration of keratinocytes and induce phagocytosis to eliminate cell debris during the inflammatory phase of wound repair [59].

To support the metabolic needs of the highly proliferative healing phase, blood vessels are essential, supporting cells with nutrition and oxygen. Both angiogenesis (sprouting of capillaries from existing blood vessels) and vasculogenesis (mobilization of bone marrow—derived endothelial progenitors) are essential to tissue repair [60].

Chronic wounds exhibit a higher expression of proteins with anti-angiogenic properties, such as myeloperoxidase, while angiogenic stimulators, such as extracellular superoxide dismutase, are generally decreased [61]. Overall, proteolytic degradation of pro-angiogenic factors, such as members of the vascular endothelial growth factor (VEGF) family, and a subsequent decrease of their bioactivity in the chronic wound microenvironment, has been suggested to be responsible for impaired tissue healing [62,63]. ELF-EMFs have been shown to influence cell migration and proliferation, and regulate mitogen activated protein kinase (MAPK) and extracellular signal-regulated kinase (ERK), promoting the following angiogenic process. In vitro and in vivo experiments revealed that ELF-EMFs impact the modulation of VEGF-dependent signal transduction pathways and increase the number of angiogenesis factors [64,65,66,67]. Furthermore, Fan et al. showed that ELF-EMFs may increase the proliferation of Human Bone Marrow Stromal Osteoprogenitor Cells (hBM-SCs), promoting DNA synthesis and increasing the proportion of cells in the S phase, and up-regulating the expressions of hematopoietic growth factors both in hBM-SCs and mesenchymal stem cells (MSC) [68]. The regulation of the differentiation and proliferation processes, necessary for optimal healing of bone fractures, is linked to the production of cytokines and growth factors. ELF-EMFs are capable of up-regulate the expressions of growth factors involved in the regenerative process, such as the macrophage colony-stimulating factor (M-CSF), stem cell factor (SCF), thrombopoietin (TPO), fibroblast growth factor 2 (FGF2), and VEGF [68,69]. Overall, changes in the wound microenvironment, with the production of several growth factors, are increased by ELF-EMF exposure, as reported by several results. These data support that the ELF-EMF exposure can be a valuable insight for angiogenesis during normal tissue repair.

### 2.4. ELF-EMFs and Remodeling Phase

Wound closure is considered the tissue injury endpoint. However, remodeling or tissue maturation can last several months or even years. This last phase of wound healing consists of a regression of the neovasculature, accompanied by deposition to the ECM and subsequent granulation tissue modification into scar tissue. In the physiological remodeling phase, collagen I synthesis, collagen III lysis, and reorganization of the ECM all take place [70]. The effect of ELF-EMFs exposure in the re-modelling phase is responsible for a change in the biological needs of the tissue. The EMF effects on collagen synthesis depend on factors including frequency, level of magnetic induction, cell density, and type. Studies reported how the application of 50 or 60 Hz ranges can modulate collagen synthesis, according to the type of skin damage and the cells involved in the remodeling mechanism [71,72]. As reported in animal models, ELF-EMFs were capable to promote healing of skin ulcers through the increase of skin collagen synthesis, but not in heat-damaged skin tissue or after skin fibroblasts activation [71,72]. The overview of experimental studies provided rational support of ELF-EMF involvement in all wound healing steps. Several studies have addressed that the ELF-EMF is differently efficient within the frequency and the intensity window, as well as the exposure time and clinical conditions, such as soft and hard tissue injury.

## 3. ELF-EMFs and Non-Healing Conditions

The onset of chronic inflammatory processes in pathological conditions, such as diabetes, hypertension, and arteriosclerosis, which may favor the onset of arterial and venous ulcers of the lower limbs. Pathological conditions may act through the induction of an insufficient blood supply, anoxia, edema, cell death, and infection, resulting in an alteration of the balance between the structural components of the affected tissues and the immune cells, which prevents wound healing [72]. Infection and alteration in biofilm composition and cellular senescence are commonly considered as an exogenous and endogenous factor that can act detrimentally in the physiological process of wound healing, leading to chronic wound development [73,74,75,76] (Figure 2).

### 3.1. ELF-EMFs Effects on Infected Wounds

Biofilms are generally polymicrobial communities, associated with each other through an extracellular polymeric substance (EPS) [73,77,78]. Usually, biofilms developing on chronic wounds are only partially (10–20 %) composed by microorganisms while EPS represents almost 80–90% composition [73]. Biofilms cause chronic wounds to become “looped” in an inflammatory state [79,80]. The ideal treatment for infection and biofilm production would be to avoid the induction of bacterial resistance phenomena and minimize the cytotoxic effects that reduce vitality, proliferation, and migration of cells involved in the wound healing process and, therefore, decrease the cure rate [81]. In the last few decades, the unregulated use of antibiotics has led to antibiotic resistance. Antibiotic resistance is an important issue in the treatment of patients with infected wounds, with alternative methods including the EMFs exposition, which are currently being investigated as the only treatment or, in combination with antibiotics, to create a synergistic effect, called a “bioelectric” effect [82,83]. Results are still controversial [74,75,76,77,78,79,80,81,82,83,84,85,86,87,88,89]. It has been demonstrated that ELF-EMFs can either negatively [77,80,85,86] or positively [84,87,88,89] affect functional parameters (cell growth and viability) of bacteria and their antibiotic sensitivity depending on the physical parameters of the electromagnetic field (frequency and magnetic flux density), the time of the exposure, and/or the type of bacterial cells used. In their study, Bayir et al. showed that the magnetic intensity, frequency, and exposure time of ELF-EMFs modified the responses of *S. aureus* and *E. coli* in colony forming capability, with a decrease for long-time exposure [90]. Moreover, Segatore et al. demonstrated that ELF-EMFs in combination with subinhibitory concentrations of antibiotics may act as stressing factors but are not able to significantly affect the bacterial growth rate. To escape from these altered or stress-producing environments, bacteria can reverse (*P. aeruginosa*) or abolish (*E. coli*) their initial responses and seek to resume their normal level of homeostasis [91]. To date, no studies have been specifically performed directly on wound infection.

### 3.2. ELF-EMFs Effects on Senescent Cells

As an endogenous factor delaying wound healing, cellular senescence has been shown to be linked with pathological tissue repair [92]. Cellular senescence was sustained by oxidative stress with a reduction in migratory and homing ability [93]. In wound healing, many types of cells are involved in different stages, and it has been reported that fibroblasts, as the main players, become prematurely senescent. Premature aging in fibroblasts was found to be telomere-independent [94] and it has been shown that it can be reversible [95,96]. Their accumulation is critical for the exacerbation of tissue damage in a chronic inflammatory setting [97].

Senescent fibroblasts and keratinocytes secrete MMP-2, MMP-3, and MMP-9, and increase matrix proteolysis and inflammation induced by the expression of senescence-associated secretory phenotype (SASP) constituents [98]. Moreover, it has been reported that senescent keratinocytes increase the production of the anti-angiogenic factor maspin (mammary serine protease inhibitor) [99], which may be detrimental to the repair process [100,101]. Some reports found that EMF could up-regulate the mRNA expression of FGF-2 [102,103], inhibiting cellular senescence and promoting cell proliferation via a phosphoinositide 3-kinase (PI3K)/AKT-Mouse double minute 2 homolog (MDM2) signaling pathway. As mentioned above, senescence is mainly characterized by the reduction of replicative and migratory activity. An in vitro study of Huo et al. reported the possible application of non-invasive ELF-EMFs as a migratory stimulator of keratinocytes and fibroblasts and as a weakly promoter of keratinocyte proliferation [26]. The observations of Manni et al. confirmed the hypothesis that ELF-EMFs (50 Hz) may modify cell membrane morphology and interfere with the initiation of the signal cascade pathway and cellular adhesion [104]. ELF-EMF applications modify the biochemical properties of human keratinocytes (HaCaT) associated with different actin productions. In their study, Patruno et al. confirmed the proliferative effect of short 50 Hz 1mT exposure on HaCaT cells, through a significant activation of the PI3K, JNK, and ERK pathways but not with p38/MAPK activation [105]. Although the ratio of non-senescent to senescent cells may influence the healing, and, in delayed tissue repair, more senescent cells were observed, the relationship between wound duration and number of senescent fibroblasts, as well as senescent keratinocytes, remains to be elucidated. Thus, it could be hypothesized that ELF-EMF application can favor the ECM synthesis, providing structural integrity and avoiding fibrosis and scarring.

### 3.3. ELF-EMFs Effects on Wound Dressing Pain

Wound dressing represents a painful experience for many patients, with distressing and disabling symptoms that have a significant effect on the Quality of Life (QoL) [16]. Pain is a complex subjective, perceptive phenomenon, which is influenced by numerous physiological, emotional, and social factors. Overall, pain has a negative effect on patient compliance and is an independent risk factor for delayed wound healing. For patients with painful wounds, it is, therefore, necessary to consider pain management as a fundamental component of care. Traditionally, pain has been divided into two categories “Nociceptive Pain” and “Neuropathic Pain”. Nociceptive pain is the normal physiological response to a painful stimulus and serves as a biologic function to warn of injury. Neuropathic pain is caused by dysfunction or damage in the nervous system. This is an inappropriate response wherein damaged nerves cause signals to travel along abnormal pathways. An important component of pain is closely linked to the mechanisms of inflammation. ELF-EMFs have been shown to affect pain and inflammation by modulating G-protein coupling receptors (GPCRs), downregulating COX-2 activity, affecting the calcium/calmodulin/nitric oxide pathway and downregulating inflammatory modulators, such as TNF-α and IL-1β as well as the NF-κB [40,44,106,107,108].

However, pain is finely modulated by specific peripheral and central nociceptors of the nervous system. The immune system can communicate, through the inflammatory process, with peripheral sensory neurons to modulate pain. Opioid receptors [109,110] are expressed on peripheral sensory nerve endings, cutaneous cells, and immune cells, such as granulocytes, monocytes, macrophages, and lymphocytes. During the inflammatory response, those cells can produce the opioid peptides that bind to three different receptors (i.e., µ, δ, and ĸ), which inhibit cyclic adenosine monophosphate (cAMP) production and/or can interact with a membrane’s ion channels [111,112,113]. Inflammation of peripheral tissue leads to the increased functionality of opioid receptors on peripheral sensory neurons and to local production of endogenous opioid peptides. Opioid receptors are widely expressed in the central and peripheral nervous systems as well as in numerous non-neuronal tissues. Ross et al. showed a greater reduction of cAMP expression in cells treated with ELF-EMFs (5 Hz), suggesting the potential use of ELF-EMFs as a complementary or alternative treatment to reduce pain and inflammation, and to enhance patient QoL, without the side-effects of opiates [114]. Furthermore, EMFs have been reported to reduce hyperalgesia and pain favoring the release of endogenous opioids into the central nervous system (CNS) [115,116,117,118,119,120]. Specifically, ELF-EMFs at 50 Hz can activate delta-opioid receptors in the spinal and rostral ventral medulla (RVM) [121].

## 4. Conclusions

The recovery process involves proliferation and migration of various cell types (epidermal, dermal, and inflammatory cells), chemical mediators’ production and the surrounding extracellular matrix organization, resulting in a tightly orchestrated re-establishment of tissue integrity. The physiological restoration of a normal tissue after injury is also subject to alteration by external and endogenous factors, such as bacterial colonization and biofilm formation, cellular senescence, and pain development.

Several experimental results support the use of magnetic fields to aid or restore wound healing interfering with all the phases of the process. An interaction between magnetic fields and healing skin has been shown to effectively modulate inflammation, protease matrix rearrangement, neo-angiogenesis, senescence, stem-cell proliferation, and re-epithelialization.

Over the past decade, considerable insights into the molecular pathways driving the healing response and impairment have suggested a possibility to use ELF-EMFs, providing scientific rationale for future clinical trials. Overall data suggest that different modalities of exposure should be applied according to chronic wounds. Since a classical treatment of chronic ulcers involves different dressings based on the diverse stages of the ulcer, in the same way, we believe, exposure to different types of magnetic fields can physiologically guide the wound healing process.

To date, there are no in vivo studies demonstrating the effects of ELF-EMFs treatment, but numerous data have been produced supporting the therapeutic role of ELF-EMFs in wound healing with in vitro and ex vivo studies, demonstrating the only potential application on immune response regulation, skin cells growing, migration etc., given the lack of confirmation of the harmful effects on health if administered in controlled ways and controlled times [122].

Until the present day, great attention for clinical/therapeutic applications was directed to pulsed fields with a greater frequency range, as compared to ELF-EMFs fields. However, scientific evidence is still limited. The ELF-EMF drives the production of specific biochemical mediators, involved in several tissue injury repairs, such as in skin, bone, and traumatic brain injury, and may represent a non-pharmacological, non-invasive, and new therapeutic application.

## Figures and Tables

**Figure 1 ijms-22-05037-f001:**
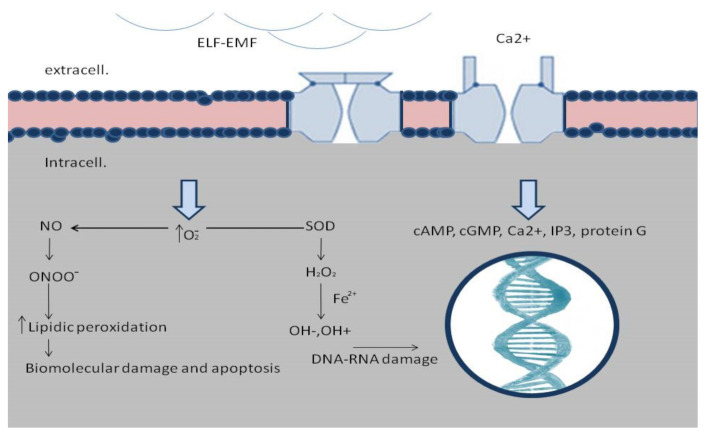
Molecular mechanisms of ELF-EMFs’ effects on cell function. ELF-EMFs open voltage-dependent calcium channels, causing interference in cell differentiation with Ca^2+^ influx into cells. It is well documented that Ca^2+^ ions affect activity-dependent gene expression, and this effect is mediated by signaling pathways activating Ca^2+^-responsive DNA regulatory elements. Decreasing antioxidants concentration has a defense mechanism against free radicals. The ELF-EMFs could also induce the production of oxygen (O_2_) in the cellular environment, which plays a major role in oxidative damage that, subsequently, led to biomolecular damage, DNA double strand breaks, DNA/RNA damage, and cell death.

**Figure 2 ijms-22-05037-f002:**
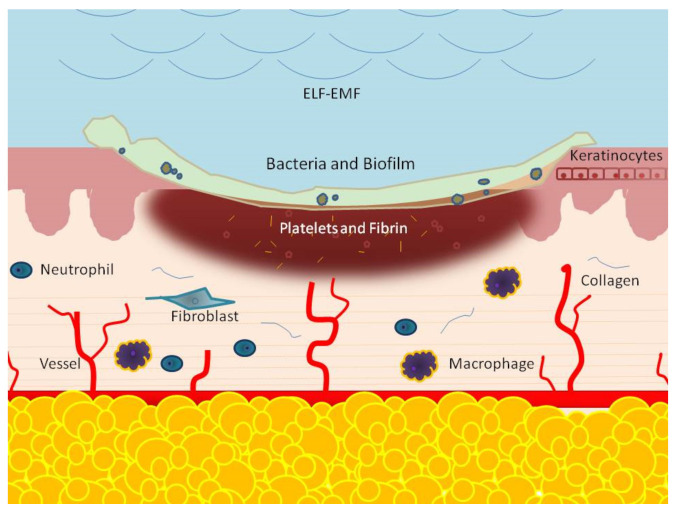
The chronic wound shows the presence of infection and biofilm formation, a hyperproliferative and nonmigratory epidermis, and an inflammatory state with an increase in inflammatory cells (neutrophils and macrophages) not properly functioning. Fibroblasts and keratinocytes become senescent while there is a reduction of angiogenesis, stem cell recruitment and activation, and ECM remodeling. ELF-EMFs has been shown to regulate the inflammatory response, induce senescence of fibroblasts, and keratinocytes through increased proliferation and migration. The regulation of MMP and collagen synthesis improves the ECM microenvironment. Proangiogenic and vasculogenic activity support cells with nutrition and oxygen. The role on biofilm and infection is still controversial.

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
