# Peer review of "Wound Repair and Extremely Low Frequency-Electromagnetic Field: Insight from In Vitro Study and Potential Clinical Application"

_ijms, 2021, doi:10.3390/ijms22095037_

Round 1

Reviewer 1 Report

The ms by Gualdi et al may represent an important element in the field of wound repair.

As minor comments, I suggest to check the english.

Author Response

We thank you for your time in reviewing the manuscript. English language was edited from an English native speaker.

Reviewer 2 Report

Dear Editor,

The manuscript summarizes the wound healing process and reviewed the findings of the effect of extremely low frequency electromagnetic fields on wound repair. The review is interesting and well-organized. It is divided in appropriate sections to facilitate understanding and follow the common thread. However, several issues should be addressed before considering the manuscript for publication:

  1. I recommend to use the full name for ELF-EMF in the title, not only the acronym.
  2. The paragraph that begins on line 56 should include the frequencies considered extremely low (300 Hz).
  3. What is the meaning of too cell migration on line 72?
  4. In several parts of the manuscript, the effect of ELF-EMF on cell behaviour or molecule secretion is detailed, although the specific details of EMFs are missing. Due to the importance of time of exposure, waveform, amplitude and other parameters on cell response, I recommend to add more information. For example on line 103 reference 35, or line 155.
  5. The increase or decrease of pro-inflammatory cytokines is a controversial issue in wound healing. In addition, in paragraph starting on line 165 is described that ELF-EMF induces anti-inflammatory cytokines, but also increase the production of pro-inflammatory cytokines (line 169). Could the authors explain this point?
  6. In section 2.3, the authors explain that ELF-EMF increases the expression of MMP-2, but reduces the collagen I synthesis. However, in section 2.4, the authors described that ELF-EMF is involved in collagen synthesis. Could the authors address this contradiction?
  7. Senescent fibroblasts and keratinocytes secrete MMP-2 and 9. How can this section influence wound healing? In section 2.3, an early up-regulation of MMP-9 induced by ELF-EMF could improve wound healing, what about senescent fibroblasts and keratinocytes secretion?
  8. The conclusion is too long, I recommend to summarize the main findings.
  9. The manuscript is well-organized in sections according the wound healing phase and the specific effect of ELF-EMF. However, there is a lack of discussion among manuscript sections. I recommend to add more discussion trying to relate the references presented in different sections. In addition, the authors should include how the recent research could be applied in clinics.  

Round 2

Reviewer 2 Report

The authors have replied all my comments and improved the manuscript accordingly. The review manuscript is ready to be accepted for publication.  

This manuscript is a resubmission of an earlier submission. The following is a list of the peer review reports and author responses from that submission.

Round 1

Reviewer 1 Report

Giulio Gualdi, Erica Costantini, Marcella Reale and Paolo Amerio reported a review manuscript entitled, “Wound Repair and ELF-EMF: clinical perspective” to International Journal of Molecular Sciences.

In abstract, the phrase “extremely low frequency electromagnetic fields (EMFs)” should be changed to “extremely low frequency electromagnetic fields (ELF-EMFs)”

Throughout the manuscript, the duration of the effect of either ELF-EMFs or EMFs should be clarified.

The limitation factors and adverse effects should be delineated in each specific representative application.

Targeting cells, tissue, their interaction and the synergism are suggested to report on ELF-EMFs.

Impact of the ELF-EMFs, compared to other modalities, should be calibrated and compared to and should explain more in detail.

Combined effect with ELF-EMFs should be elucidated if any is present.

While the title of the manuscript contains “clinical”, there are les clinical indications or implications in the manuscript. More clinical relevance and effects should be included.

Author Response

1) we replaced the terms as required and corrected english language

2) Regarding this topic, unfortunately over half of cited published works do not report the exposure time. Initially it was our intention to produce a summary table but because the lack of such data and low uniformity of the exposure times, we believe would have produced confusion. if the reviewer deems it necessary we can produce the table

3)4)5)6) we added in "conclusion":  "The major limitation is that, to date, there are no in vivo studies  and that therefore adverse events are only potential (immune dysregulation could alter response to potential pathogens, potential dysregulation of stem cells growing, etc.) Furthermore importance of clinical study is pivotal since even several earlier assumption on the proposed effects of ELF-EMF, such as the allegation of brain tumor development, have recently been reversed. As no in vivo study have been performed with ELF-EMF it is difficult to compare this method with others, however there is a potential advantage in a method, such the one here described, that may interfere with so many element of wound-healing process at the same time, without a direct interaction with the tissue. However of high importance is the lack of standardized nomenclature and experimental protocols when it comes to running trials. Like pharmaceuticals, where dosages vary from patient to patient, EMF treatments also are dose and tissue dependent. Furthermore, no studies have ever addressed the interaction between biological systems or different tissues under the influence of ELF-EMF, so no conclusions can be drawn. Finally, the combination of ELF-EMF with other substances has been variously studied in areas other than wound healingand it is also known that the MRI contrast agent can increase its cytotoxic effect on cells after irradiation with ELF- ELM and therefore also other interactions with agents could induce the same effect . In the wound healing area attempts have been made  to combine ELF-EMF with metallic nanoparticles without any additional effects on proliferation."

These considerations have been added to the discussion with related references.

7) As there are no clinical studies, in accordance with the suggestion, we change the title to "Wound Repair and ELF-EMF: insight from in vitro study and potential clinical application"

Reviewer 2 Report

The ms by Gualdi et al may represent an important element in the field of wound repair.

As minor comments, I suggest to put some details about negative effects of EMF on wound repair, as well as to cite relevant clinical applications.

Author Response

We added in "conclusion":  "The major limitation is that, to date, there are no in vivo studies  and that therefore adverse events are only potential (immune dysregulation could alter response to potential pathogens, potential dysregulation of stem cells growing, etc.) Furthermore importance of clinical study is pivotal since even several earlier assumption on the proposed effects of ELF-EMF, such as the allegation of brain tumor development, have recently been reversed. As no in vivo study have been performed with ELF-EMF it is difficult to compare this method with others, however there is a potential advantage in a method, such the one here described, that may interfere with so many element of wound-healing process at the same time, without a direct interaction with the tissue. However of high importance is the lack of standardized nomenclature and experimental protocols when it comes to running trials. Like pharmaceuticals, where dosages vary from patient to patient, EMF treatments also are dose and tissue dependent. Furthermore, no studies have ever addressed the interaction between biological systems or different tissues under the influence of ELF-EMF, so no conclusions can be drawn. Finally, the combination of ELF-EMF with other substances has been variously studied in areas other than wound healingand it is also known that the MRI contrast agent can increase its cytotoxic effect on cells after irradiation with ELF- ELM and therefore also other interactions with agents could induce the same effect . In the wound healing area attempts have been made  to combine ELF-EMF with metallic nanoparticles without any additional effects on proliferation."

These considerations have been added to the discussion with related references.

 As there are no clinical studies, in accordance with the suggestion, we change the title to "Wound Repair and ELF-EMF: insight from in vitro study and potential clinical application"

Round 2

Reviewer 1 Report

Giulio Gualdi, Erica Costantini, Marcella Reale and Paolo Amerio reported a review manuscript entitled, “Wound Repair and ELF-EMF: insight from in vitro study and potential clinical application” to International Journal of Molecular Sciences.

Since there are few in vitro studies and no in vivo study and then no good clinical trial or study, this manuscript should focus more on “potential” mechanism of cellular, molecular, genetic and epigenetic issues.

Current format of this manuscript is merely explaining how ELF-EMF functions in cells, inter-cellular fashion or in between tissues.

The authors should pick up more “wound healing” relevant such events, i.e., in terms of each wound healing phase, what kind of mode of action is it taken in various wounding or how these may be regulated and interacted in such conditions.